# Machine learning-based prediction of in-hospital mortality using admission laboratory data: A retrospective, single-site study using electronic health record data

**Tomohisa Seki**[1]*, **Yoshimasa Kawazoe**[1,2], **Kazuhiko Ohe**[1,3]

1 Department of Healthcare Information Management, The University of Tokyo Hospital, Tokyo, Japan,
2 Artificial Intelligence in Healthcare, Graduate School of Medicine, The University of Tokyo, Tokyo, Japan,
3 Department of Medical Informatics and Economics, Graduate School of Social Medicine, The University of Tokyo, Tokyo, Japan

* seki@m.u-tokyo.ac.jp

**Data Availability Statement:** The data used in this study were not openly available due to the restriction imposed by the research ethics committee of the Graduate School of Medicine and

## Abstract

Risk assessment of in-hospital mortality of patients at the time of hospitalization is necessary for determining the scale of required medical resources for the patient depending on the patient's severity. Because recent machine learning application in the clinical area has been shown to enhance prediction ability, applying this technique to this issue can lead to an accurate prediction model for in-hospital mortality prediction. In this study, we aimed to generate an accurate prediction model of in-hospital mortality using machine learning techniques. Patients 18 years of age or older admitted to the University of Tokyo Hospital between January 1, 2009 and December 26, 2017 were used in this study. The data were divided into a training/validation data set (n = 119,160) and a test data set (n = 33,970) according to the time of admission. The prediction target of the model was the in-hospital mortality within 14 days. To generate the prediction model, 25 variables (age, sex, 21 laboratory test items, length of stay, and mortality) were used to predict in-hospital mortality. Logistic regression, random forests, multilayer perceptron, and gradient boost decision trees were performed to generate the prediction models. To evaluate the prediction capability of the model, the model was tested using a test data set. Mean probabilities obtained from trained models with five-fold cross-validation were used to calculate the area under the receiver operating characteristic (AUROC) curve. In a test stage using the test data set, prediction models of in-hospital mortality within 14 days showed AUROC values of 0.936, 0.942, 0.942, and 0.938 for logistic regression, random forests, multilayer perceptron, and gradient boosting decision trees, respectively. Machine learning-based prediction of short-term in-hospital mortality using admission laboratory data showed outstanding prediction capability and, therefore, has the potential to be useful for the risk assessment of patients at the time of hospitalization.

Faculty of Medicine, The University of Tokyo, because they contain sensitive patient information. Disclosure of data is not included in the ethics application, nor is it allowed in this study by hospital policy. Specific contact information is as follows: Office for Human Research Studies (OHRS): Graduate School of Medicine and Faculty of Medicine, The University of Tokyo, Faculty of Medicine Bldg. 2 4F, 7-3-1, Hongo, Bunkyo-ku, Tokyo, 113-0033, JAPAN, https://u-tokyo-ohrs.jp/en/.

**Funding:** This research was supported by the Tokyo Society of Medical Sciences and Grant-in-Aid for Scientific Research (C). There was no additional external funding received for this study.

**Competing interests:** Y.K belongs to the "Artificial Intelligence in Healthcare, Graduate School of Medicine, The University of Tokyo" which is an endowment department, supported by an unrestricted grant from "I&H Co., Ltd." and "EM SYSTEMS company." However, these sponsors had no control over the interpretation, writing, or publication of this work. This does not alter our adherence to PLOS ONE policies on sharing data and materials.

## Introduction

Mortality prediction models of patients have long been developed to objectively assess the severity of the patients and share it among the medical care team to achieve collaborative care [1, 2]. Additionally, the accurate mortality risk assessment of patients at the time of hospitalization is necessary for determining the scale of required medical resources needed according to the patient's severity. While some mortality prediction systems such as APACHE score [3–6], SAPS score [7–9], and SOFA score [10] focused on the cases of adult intensive care unit admissions, the model of mortality prediction for the cases of overall hospitalization have been limited. To determine the scale of required medical resources according to the patient's severity at the time of hospitalization, the mortality prediction model that covers the overall hospitalized population is required.

To generate the mortality prediction model, laboratory values have been known to be useful in achieving favorable prediction capability [11–14]. Additionally, in contrast to the checking of vital signs, laboratory tests are hardly repeated in a short amount of time, thus, it is easy for clinicians to decide which measured laboratory values to input in the prediction model. Therefore, results of laboratory tests are relatively manageable to input in the prediction model. Recent machine learning applications using nonlinear feature extraction in the clinical setting have been shown to enhance prediction ability [15, 16], and applying these types of techniques to this issue can lead to generating an accurate prediction model for in-hospital mortality prediction. Therefore, in this study, we generated machine learning models of in-hospital mortality prediction with nonlinear feature extraction using laboratory data, compared them with a logistic regression model that uses linear feature extraction, and tried to generate an accurate prediction model of in-hospital mortality at the time of hospitalization.

## Materials and methods

### Study population and data sources

Data of patients who were admitted to the University of Tokyo Hospital between January 1, 2009 and December 26, 2017 were used in this study. The study was approved by the research ethics committee of the Graduate School of Medicine and Faculty of Medicine of the University of Tokyo. At this time, the data used in this study was not openly available due to the restriction imposed by the research ethics committee (Office for Human Research Studies (OHRS); Graduate School of Medicine and Faculty of Medicine, The University of Tokyo, Faculty of Medicine Bldg. 2 4F, 7-3-1, Hongo, Bunkyo-ku, Tokyo, 113–0033, JAPAN, https://u-tokyo-ohrs.jp/en/). For this study, the total number of patients was 80,729, and the total number of admission cases was 173,578. Admission laboratory data were defined as the data of the sample collection times closest to that of the hospitalization decision order within six hours before and after the hospitalization decision order was given. These data were extracted from HL7 message documents maintained in de-identified SS-MIX2 standardized storage [17] using Python (version 3.6.8) scripts with respect to each admission case. After data extraction, admission data with transcription errors of the time of admission were excluded. The data were then divided into a training/validation data set and a test data set depending on the time of admission (training/validation data set: n = 134,961, between January 1, 2009 and December 31, 2015; test data set: n = 38,616, between January 1, 2016 and December 26, 2017). Admission cases under 18 years of age were excluded from both data sets, while the admission cases of those 18 years of age or older were defined as eligible cases for the analysis as shown in Fig 1. Variables with > 25% missing data or variables not suitable for developing the prediction model in the training/validation set were also excluded from the analysis. The anonymous

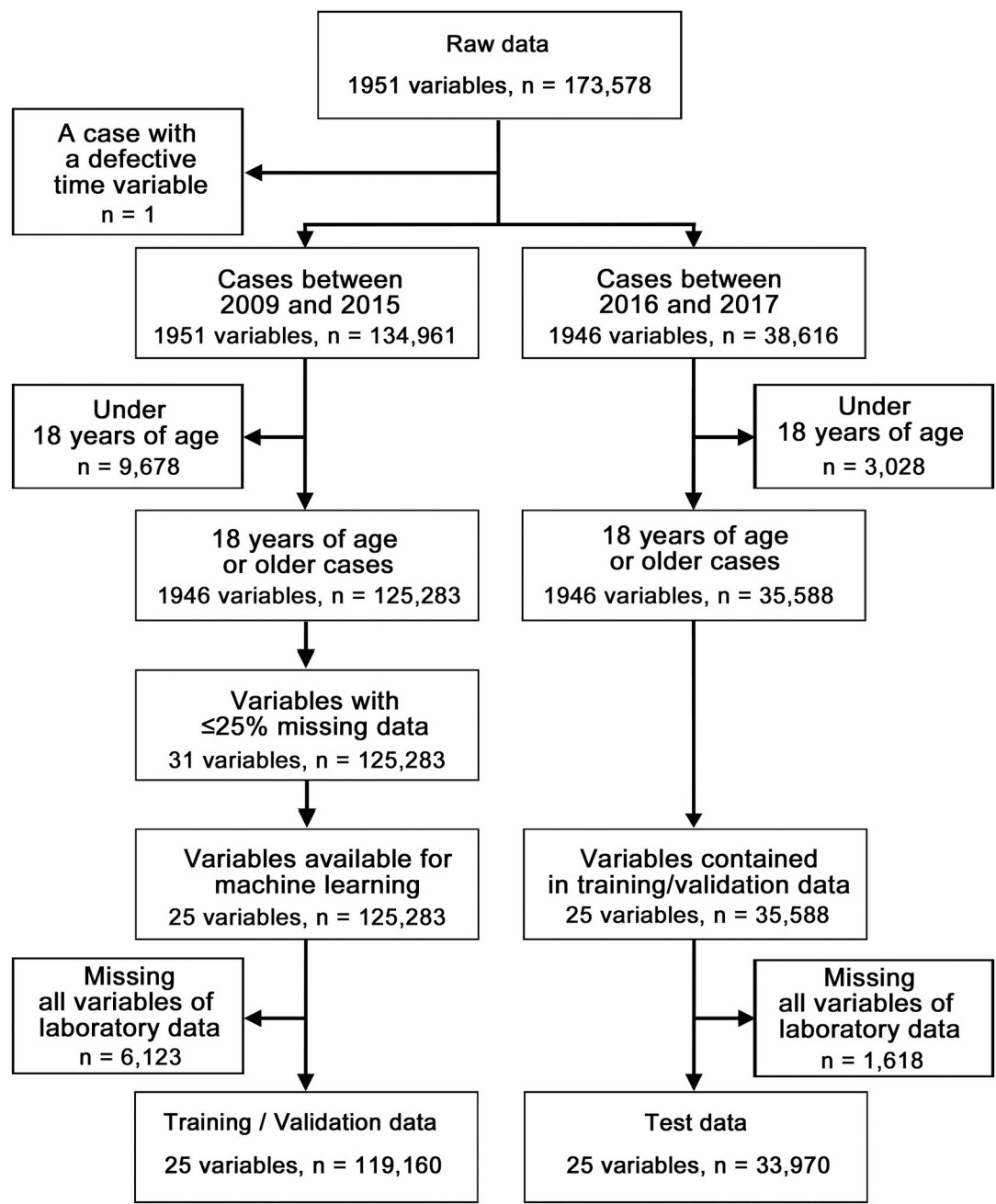

**Fig 1. Schema of selection of admission cases and data preprocessing.**

patient's ID, hospitalization time, and alkaline phosphatase value displayed in King-Armstrong unit were excluded from the remaining variables. Because alkaline phosphatase values in IU/l were included in the data, we removed the values with King-Armstrong unit. Ultimately, 25 variables, specifically, age, sex, 21 laboratory variables, length of stay, and mortality, were considered eligible for analysis. Subsequently, cases that were missing all variables of eligible laboratory data were excluded. Finally, a training/validation data set (n = 119,160) and a test data set (n = 33,970) were obtained.

## Imputation of missing data

In the training/validation data set, 43,535 of 119,160 records had missing data while 11,332 of 33,970 records had missing data in the test set. Missing data were imputed 20 times with probability-based multiple imputation using the micemd package (version 1.2.0) with R (version 3.3.2). The imputation method for each variable was selected by an internal algorithm of the micemd package according to the structure of the incomplete dataset following previously published guidelines [18]. To draw the receiver operating characteristic (ROC) curve and the precision-recall curve, the results of the imputation and the bootstrapped confidence interval (CI) based on 2000 replications were used with majority voting by aggregating the results of the imputation. To evaluate the distribution of the predicted probability of in-hospital mortality and observed in-hospital mortality, the prediction results were collected based on majority voting by aggregating the results of the imputation.

## Machine learning

To generate prediction models of in-hospital mortality, four supervised machine learning methods (logistic regression [19], random forests [20], multilayer perceptron [21], and gradient boosting decision trees [22]) were applied to the training data set, respectively. The variables used for model training were 23 variables (age, sex, and 21 laboratory items on admission). The outcome as the prediction target was in-hospital death within 14 days. To cope with class imbalance data, the synthetic minority over-sampling technique (SMOTE) algorithm [23] was applied to artificially increase the rate of cases of in-hospital mortality of the training data to 20% of the number of all cases in training data. Additionally, the class-balanced loss was used in the training phase of machine learning. To train the models and optimize the hyper parameters of the model, all of the imputed training data were used. The hyper parameters which maximized the average values of AUROC obtained from 5-fold cross-validation within the training data were selected. In the 5-fold cross-validation, the training/validation data were split into the training data and the validation data. However, distributing the imputed cases derived from an identical admission case to both of the training data and the validation data was avoided.

In logistic regression, the hyper parameters of the regularization coefficient ($10^2$ to $10^4$) and the regularization method (L1 or L2) were tested using grid search. In the random forests, under fixing the number of decision trees to 500, the maximum depth of each decision tree (1 to 20) and the maximum number of explanatory variables (1 to 20) used for classification in the decision nodes were tested using a grid search to optimize the model. In the multilayer perceptron, a three-layer neural network in which each layer had 128 perceptrons was applied using the stochastic gradient descent method as an optimization method. For the hyper parameter optimization, the dropout ratio of perceptrons (0 to 0.9), the learning rate ($10^2$ to $10^3$), and the batch size (128 to 2048) were tested using grid search. In the gradient boosting decision trees, the number of leaves (10 to 400) and the minimum number of data in leaves (10 to 400) were tested using a grid search to optimize the model. Validation data were used to estimate the termination point of training using the early stopping protocol in the multilayer perceptron method. These analyses of the machine learning stage were performed using Python (version 3.5.2) scripts and DGX-1 with Tesla P100.

The prediction models after training and optimizing hyper parameters were tested using all of the test data, and the AUROC was calculated for each model. Because the probability calibration for the output value of each machine learning model was performed using isotonic regression on the validation data at each phase of cross-validation to imitate evaluation of predictive performance for future data, the average value of the output values of the five models

generated by cross-validation was used for calculating the final AUROC. Shapley additive explanations (SHAP) values were calculated with SHAP module of Python [24] (https://github.com/slundberg/shap).

## Results

Descriptive statistics values of the variables in the training/validation data and the test data after the data preprocessing are shown in S1 Table. The proportions of in-hospital mortality within 14 days were 0.84% for training/validation data and 0.69% for test data (S1 Table). Year-by-year mortality is shown in S1 Fig.

In the development phase of the prediction model, the models were trained to predict in-hospital mortality within 14 days using 23 variables (age, sex, 21 blood sampling items) on the admission (Fig 2). After model training, the AUROC was calculated from the results of predicting in-hospital mortality using 23 items with trained machine learning models against test data. As a result, the AUROC values of 0.936 [95% CI, 0.920–0.950], 0.942 [95% CI, 0.929–0.954], 0.942 [95% CI, 0.928–0.954], and 0.938 [95% CI, 0.923–0.952] were obtained for logistic regression, random forests, multilayer perceptron, and gradient boosting decision tree, respectively (Fig 3A). These results indicate that the models generated with machine learning methods using admission laboratory data achieve favorable prediction capability to predict short-term in-hospital mortality.

Because the prediction task used highly imbalanced data, area under the precision-recall curve (AUPRC) was calculated in each machine learning method. For a detailed comparison of prediction models in binary classification using imbalanced data, the AUPRC is more informative than the AUROC [25]. As a result, the prediction model derived from the gradient boosting decision tree showed the highest value of AUPRC, though these models showed a small difference in AUROC values (Fig 3B). Additionally, the distribution of predicted probability of in-hospital mortality and observed in-hospital mortality showed that the mortality risk of high-risk patients tended to be predicted accurately in the case of the gradient boosting decision tree (Fig 3C, S2 Fig). These results indicate that our prediction model generated with the gradient boosting decision tree has the possibility to be practically useful compared to the other three prediction models for extracting high-risk patients from a population that is largely comprised of low-risk patients.

To evaluate the calibration of the prediction model, the in-hospital mortality prediction probability was stratified assuming actual clinical use and was compared to observed in-hospital mortality. As a result, the in-hospital mortality rates in each stratified group fell within the range of prediction probability as shown in Fig 3C. These results indicate that this prediction model has the possibility to be applied to accurate stratification of short-term in-hospital mortality risk.

To investigate the importance of variables and the validity of the internal algorithm of the prediction model, SHAP values were calculated in cases of all four prediction models. Assigning each variable an importance value for a particular prediction, variable importance was visualized in each case of the test data sets. As a result, although rank of variable importance differed depending on machine learning methods, a few variables such as serum lactate dehydrogenase (LDH) and serum albumin were consistently ranked high (Fig 4). Because LDH and serum albumin have been previously reported as markers for predicting prognosis in hospitalized patients [26, 27], this result indicates that internal algorithms of the prediction models are consistent with previous evidence.

## Discussion

In this study, we successfully generated a prediction model of in-hospital mortality within 14 days using the data stored in electronic health records of the university hospital in Japan. In

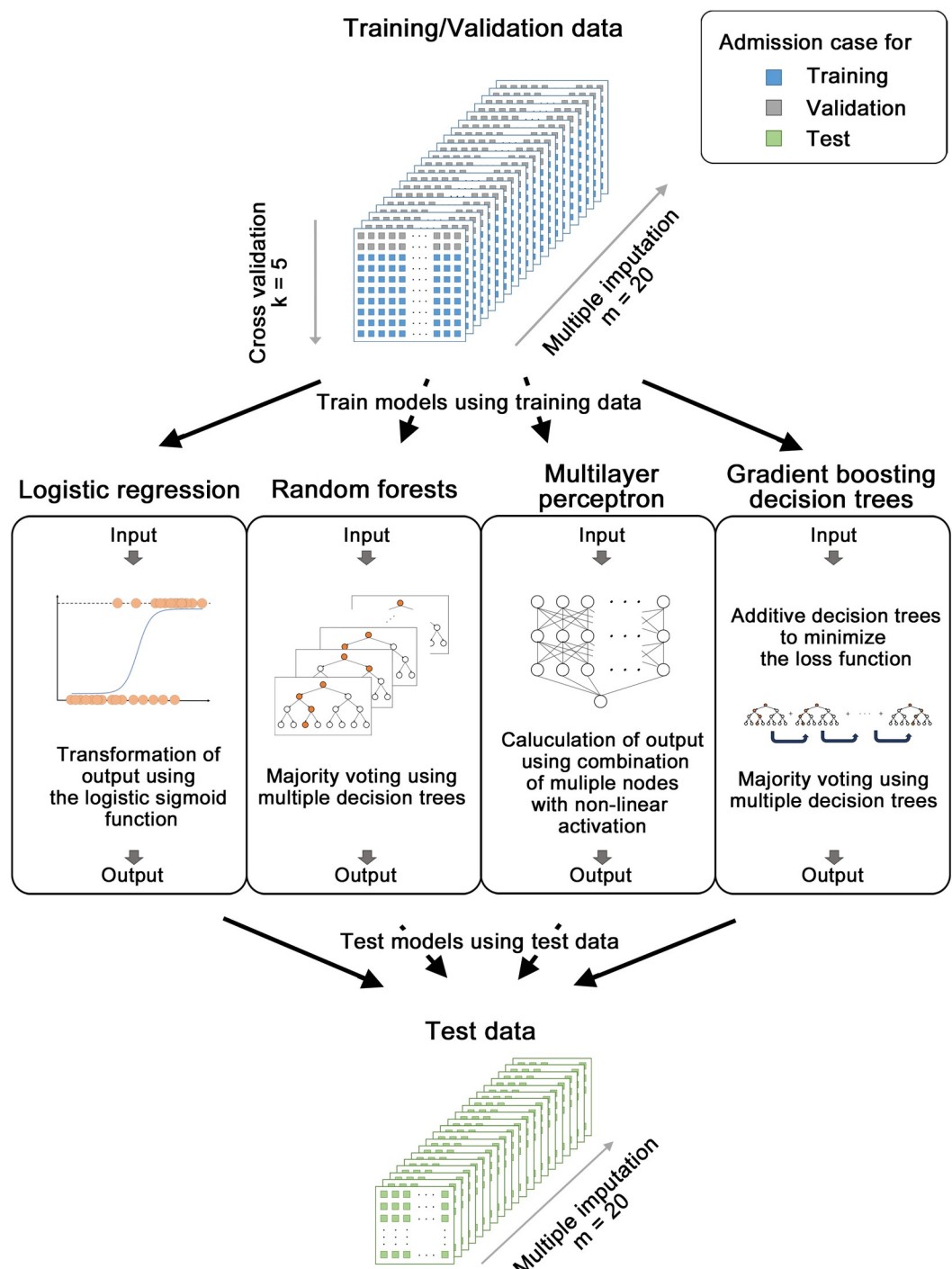

**Fig 2. Scheme of multiple imputation, cross-validation, training the model, and testing the model.** The missing data are filled with multiple imputation in m (= 20) times. As a result, m (= 20) complete data sets were generated after multiple imputation. In the training phase, cross-validation was performed in the condition of k (= 5) fold. Four machine learning methods (logistic regression, random forest, multilayer perceptron, and gradient boosting decision tree) were applied in this study.

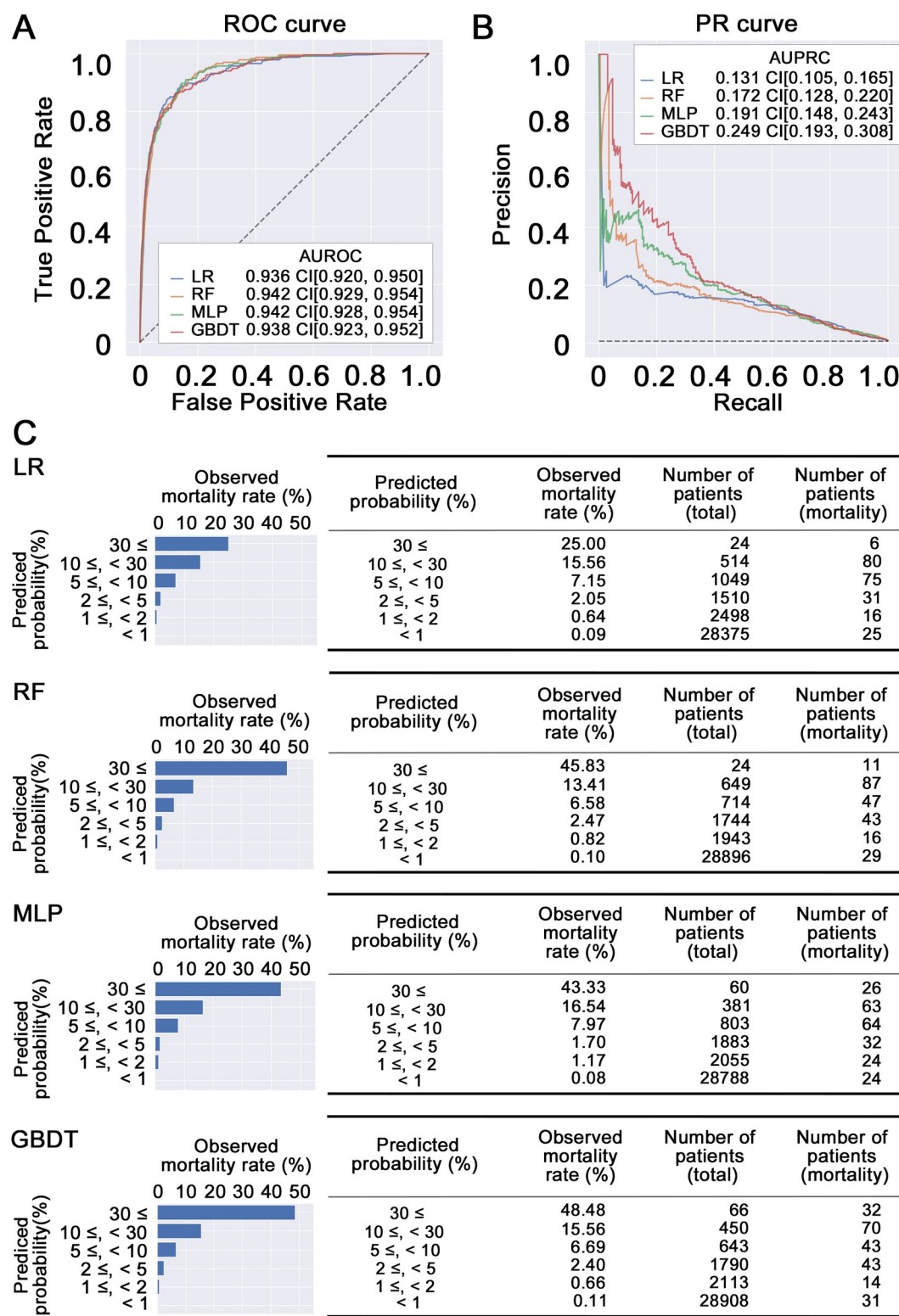

**Fig 3.** (A) The receiver operating characteristics curve and the area under the receiver operating characteristic (AUROC) curve of the models. The results of the prediction using all test data are shown. AUROC is shown with 95% confidence interval (CI). (B) The precision-recall curve and the area under the precision-recall (AUPRC) curve of the models. The results of the prediction using all test data are shown. AUPRC is shown with 95% confidence interval (CI). (C) The distribution of predicted probability of in-hospital mortality and observed in-hospital mortality. The results of logistic

regression (LR), random forest (RF), multilayer perceptron (MLP), and gradient boosting decision tree (GBDT) are shown. The bar graph shows the distribution of in-hospital mortality of test data (n = 33,970) and predicted probabilities of in-hospital mortality obtained from the prediction models (left). The table shows the observed in-hospital mortality and detail number of patients at each range of predicted probabilities obtained from the prediction models (right).

our examination, prediction of in-hospital mortality with 23 variables (age, sex, 21 blood sampling items) at the time of admission using machine learning models showed favorable values for AUROC and capability of risk stratification. Although in-hospital mortality prediction models based on logistic regression using blood test results at the time of admission have been previously reported [12–14, 28], the study of comparative evaluation of the machine learning method with linear feature extraction and nonlinear feature extraction has been limited. The sufficient amount of data used in this study, the machine learning methods which were capable of dealing with nonlinear classification, and the fine optimization of the hyper parameters have the possibility to contribute to achieving high prediction performance of our models.

One important limitation of this study was that, though the models demonstrated prediction capability, our results did not necessarily guarantee that laboratory data accounted for the majority of the prediction capability in this prediction task. Unstructured information outside SS-MIX2 standardized storage may show better results if it included explanatory variables after data structuring. In fact, an attempt using a deep learning model on an entire raw electronic health record dataset has been recently made, and it demonstrated preferable prediction capability [29]. To clarify whether the laboratory data provide the majority of capability in predicting in-hospital mortality, further investigation is needed.

Additionally, this study currently has some important limitations in terms of predictive performance. First, only data from a single facility were used in this study. It is necessary to verify whether the high prediction performance that was obtained in this study will be reproduced for the data of other facilities. Additionally, because this study uses data from a single university hospital, an internal algorithm of the model has the possibility to focus on features that are not common in general hospitals. For example, the data used in this study include cases of admission for palliative care. Because the treatment for this type of admission is different from common admission with the purpose of lifesaving, this heterogeneity of admission types has the potential to make the model focus on the features of patients in the terminal phase who were hospitalized for non-lifesaving purposes. Therefore, if this model is used in clinical practice, the user should pay attention to the limitation that the model was developed using data that includes cases admitted to the hospital not for non-lifesaving purposes. Furthermore, although the laboratory variables used in this study were limited to variables with $\leq$ 25% missing data and rare laboratory tests were not included, these models did not allow the use of missing values as input. In real-time clinical use, there will be cases where additional testing is required to actively avoid creating missing values. In retrospective use, these models are not applicable to few cases, such that missing value completion is not possible.

Second, since this study does not consider the diagnosis in predicting in-hospital mortality, there is a possibility that the prediction performance varies among groups of different diagnoses. Similarly, the importance of the variables in predicting in-hospital mortality using this model is not guaranteed in each diagnostic group. For example, in this study, LDH and serum albumin were found to be important variables in predicting in-hospital mortality in the variables we used for building machine learning models. In line with this, it has actually been previously reported that LDH and serum albumin were markers for predicting prognosis in hospitalized patients [26, 27], and although this indicates that internal algorithms of the prediction models are consistent with previous evidence, the results of this study do not prove the

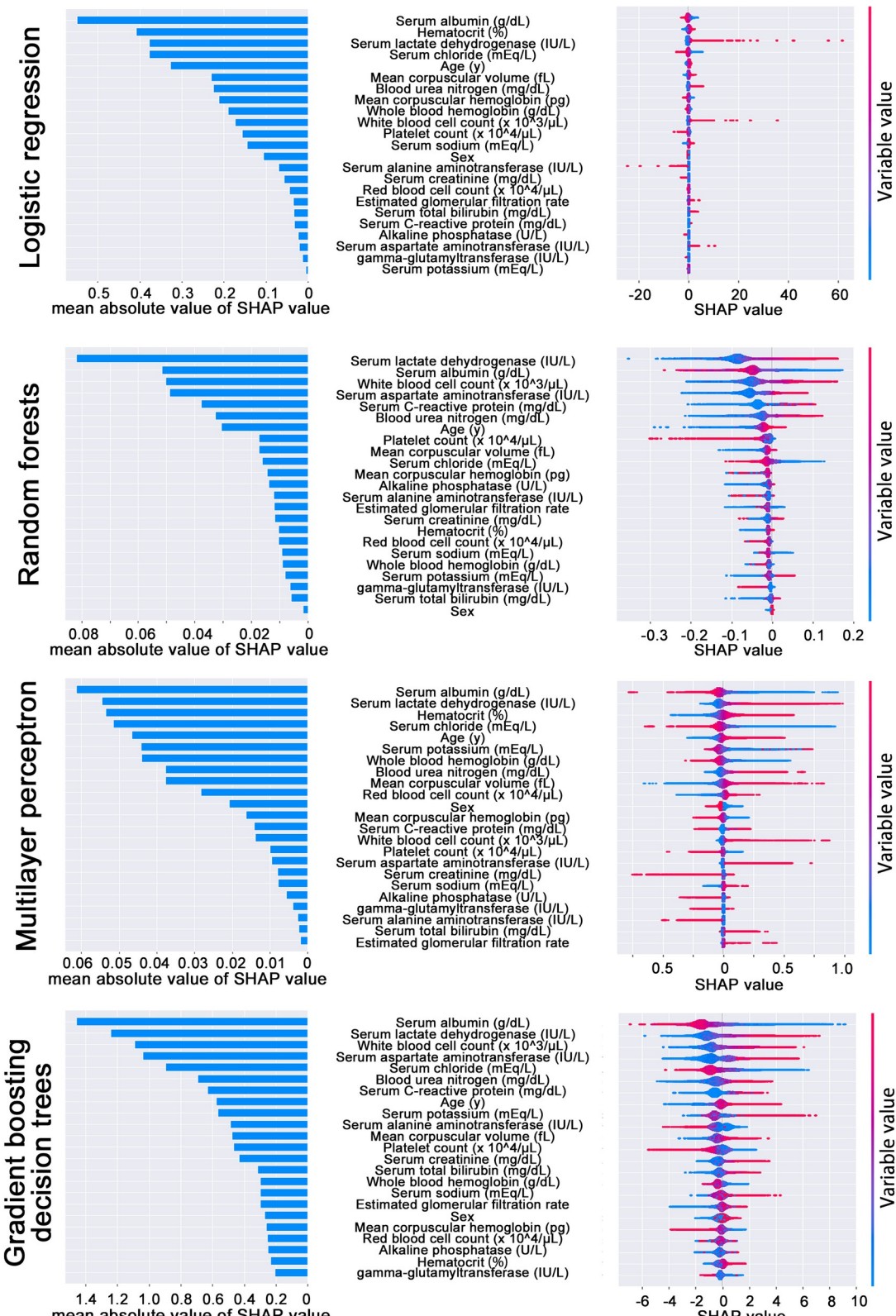

**Fig 4. The results of Shapley additive explanations (SHAP) value calculation.** The results of logistic regression, random forest, multilayer perceptron, and gradient boosting decision tree are shown. The figures on right side show the distribution of the SHAP value calculated with the test data set (n = 33,970). One plot means one prediction result in the test data set. The color

of plots shows variable values as shown in color scale bars on the right side of the figures. In the figure, a positive SHAP value means contribution to in-hospital mortality in 14 days and a negative SHAP value means the opposite. The bar graph on the left side shows the mean absolute value of SHAP value in each variable. The names of variables are displayed in the center in order of mean absolute value of SHAP value.

cause-and-effect relationship between the variables and outcome. Furthermore, the results of this study do not indicate that these markers are useful in predicting in-hospital mortality for every diagnostic group in which LDH or serum albumin has not been proven to be a prognostic predictor. Therefore, for specific diagnostic groups, it is important to prioritize risk assessment with existing prediction models that are proven to be clinically useful. For this reason, the target range of this machine learning model should be limited to cases wherein such an existing prediction model does not exist or cannot be used.

A limitation to the clinical application of the machine learning model is that the discrepancy in the event rate between the available data and future data cannot be calibrated. Actually, in this study, the mortality rate of the test data in 2016 and 2017 was comparatively low throughout the entire dataset (S1 Fig). If there is any chronological trend in the entire data, older training data may affect model performance. Therefore, ongoing model updates may be required for applying the machine learning model to clinical use.

## Conclusion

In this study, we developed a model that predicts in-hospital mortality within 14 days with high predictive performance using machine learning technology and variables of age, sex, and blood sampling test results of 21 items recorded in the electronic medical record at the time of hospitalization. This machine learning model has the possibility to be useful in evaluating the in-hospital mortality risk of admitted patients.

## Supporting information

**S1 Fig. Changes in mortality by year.** The bars show the in-hospital mortality rates for each year.
(TIF)

**S2 Fig. Calibration plot of prediction results.** On the left side of the figure, each plot figure shows the observed probability and predicted probability based on 10 quantiles of predicted probability. On the right side of the figure, each plot figure shows the observed probability and predicted probability based on 10 equal parts of predicted probability. Confidence intervals are calculated using the F distribution.
(TIF)

**S1 Table. Baseline characteristics of variables.** On the left side of the columns of each data, continuous variables are presented as the mean value and standard deviation (SD) after imputation, and categorical variables are shown as percentages after imputation. Missing rates before imputation are presented as percentages on the right side of the columns.
(DOCX)

## Acknowledgments

The authors would like to thank all of the staff and the graduate students of the Department of Healthcare Information Management at the University of Tokyo Hospital for providing an opportunity to continue this research.

## Author Contributions

**Conceptualization:** Tomohisa Seki.

**Data curation:** Tomohisa Seki.

**Formal analysis:** Tomohisa Seki.

**Funding acquisition:** Tomohisa Seki.

**Investigation:** Tomohisa Seki.

**Methodology:** Tomohisa Seki.

**Project administration:** Yoshimasa Kawazoe, Kazuhiko Ohe.

**Resources:** Kazuhiko Ohe.

**Supervision:** Yoshimasa Kawazoe.

**Validation:** Tomohisa Seki.

**Visualization:** Tomohisa Seki.

**Writing – original draft:** Tomohisa Seki.

**Writing – review & editing:** Tomohisa Seki, Yoshimasa Kawazoe, Kazuhiko Ohe.

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
