## [Decision Letter · Decision Letter 0]

10 Nov 2020

PONE-D-20-20161

Machine Learning-Based Prediction of In-hospital Mortality Using Admission Laboratory Data

PLOS ONE

Dear Dr. Seki,

Thank you for submitting your manuscript to PLOS ONE. After careful consideration, we feel that it has merit but does not fully meet PLOS ONE’s publication criteria as it currently stands. Therefore, we invite you to submit a revised version of the manuscript that addresses the points raised during the review process.

Please pay particular attention to comments provided by Reviewer 1. The differences in training and testing sets and the validation strategies required additional discussion.

We look forward to receiving your revised manuscript.

Kind regards,

Bobak Mortazavi

Academic Editor

PLOS ONE

Journal Requirements:

'This research was partially supported by the Tokyo Society of Medical Sciences and Grant-in-Aid for Scientific Research (C). The funders had no role in study design, data collection and analysis, decision to publish, or preparation of the manuscript.'

a. Please provide an amended statement that declares *all* the funding or sources of support (whether external or internal to your organization) received during this study, as detailed online in our guide for authors at http://journals.plos.org/plosone/s/submit-now

Please also include the statement “There was no additional external funding received for this study.” in your updated Funding Statement.

'I have read the journal's policy and the authors of this manuscript have the following competing interests: Y.K belongs to the 'Artificial Intelligence in Healthcare, Graduate School of Medicine, The University of Tokyo' which is an endowment department, supported with an unrestricted grant from ‘I&H Co., Ltd.’ and ‘EM SYSTEMS company’, but these sponsors had no control over the interpretation, writing, or publication of this work.'

a. Please confirm that this does not alter your adherence to all PLOS ONE policies on sharing data and materials, by including the following statement: "This does not alter our adherence to  PLOS ONE policies on sharing data and materials.” (as detailed online in our guide for authors http://journals.plos.org/plosone/s/competing-interests).  If there are restrictions on sharing of data and/or materials, please state these.

Please note that we cannot proceed with consideration of your article until this information has been declared.

5. Please include a copy of Table 1 which you refer to in your text on Line 156.

Reviewers' comments:

Reviewer's Responses to Questions

**Comments to the Author**

1. Is the manuscript technically sound, and do the data support the conclusions?

Reviewer #1: Partly

Reviewer #2: Yes

2. Has the statistical analysis been performed appropriately and rigorously? 

Reviewer #1: No

Reviewer #2: Yes

3. Have the authors made all data underlying the findings in their manuscript fully available?

Reviewer #1: No

Reviewer #2: No

4. Is the manuscript presented in an intelligible fashion and written in standard English?

Reviewer #1: Yes

Reviewer #2: Yes

5. Review Comments to the Author

Reviewer #1: The authors developed four machine learning models to predict in-hospital mortality risks. The models had similar performance in terms of discrimination and calibration. I have the following comments:

- The authors argued that they chose the lab data because it is the most manageable to input into the model. However, nowadays all data available in electronic health records can be readily input into a model that is integrated in the system. They might obtain better results if including additional data but also need to investigate whether the lab data does provide the majority of the prediction capability.

- It is not clear what they meant by "variables not suitable for developing prediction model".

- It is not mentioned in the paper, but I assume they performed multiple imputation for the test set also, judging from their figures. How did they calculate the predictions from the multiply imputed test set then? Please specify. Also, for application in a clinical setting, how do they expect the data to handle missingness?

- Why did the authors use the five models trained in training-fold of the cross-validation for application on the test set? The models can benefit more by being trained on the entire training set (or in the authors' terms, the training+validation data).

- The models were calibrated during training phase. Were the models calibrated for the test set? There is discrepancy in the event rate between the training and test sets, which may require calibration.

- For the calibration of the models, the authors need to show their results in deciles of risks or plot a smooth calibration curve. Right now the majority of patients are in the <1% risk category and it is hard to determine how accurately their risks are calibrated.

- Most importantly, the paper lacks a baseline model and has not explained the advantage of the gradient descent boosting model over logistic regression model. The authors need to compare their models with existing models on the same dataset; comparing roc-aucs on different datasets may not be fair. Also, the authors stated that the gradient descent boosting model had the best performance, but the difference from other models is small. It may be helpful to show percentage of patients who had better predictions.

- The figures have very low resolution. The words in the figures are not legible.

Reviewer #2: Seki ea al. present here a retrospective analysis of predicting mortality following a hospital stay by using several machine learning techniques in a dataset spanning nine years, 80k patients, and 174k admissions. They robustly utilized multiple imputation on missing variables, oversampled to account for outcome class imbalance, and clearly describe their approaches in grid searching model parameters. The model here uses 21 common lab results, age, and sex to predict mortality. As such, this is a model that could likely be applied widely. The authors note that a key limitation is that this model at present has only been applied to patients from a single hospital. I hope that in future work, this model is able to be tested and deployed more widely to aid in clinical care.

One concern I have in reading this manuscript is that the mortality in the test set is so much lower than in the train set (0.69 vs 0.84). While the analysis and metrics used are not substantially impacted by this imbalance, it gives me concern that the model might not be optimally calibrated. What does year-by-year mortality look like in this dataset? Has it been increasing across the entire dataset, or is the decrease present only in the two years of the training set? If there are changes over time in the dataset, what impact would there be from removing an older year of training data? I realize that not all of these questions can be answered, but it would be good to see the authors acknowledge and discuss some of the implications of this change over time.

Throughout the paper, the authors reference different numbers of variables- either 25 or 23. I believe that the authors were uniform in their approach and that the 24th and 25th variables are length of stay and mortality, and so are not included as features. However, discussion such as in line 98 or in Figure 1 implies (perhaps misleadingly) that there were 25 features in the dataset. Clarification here would be useful.

On lines 96-97, the authors note that variables "not suitable for developing the prediction model . . . were excluded." It would be helpful if the authors could clarify how these variables were unsuitable- for instance, is it because they were personal identification variables? Free text variables? Some other reason? As it appears from Figure 1 that 6 variables fall into this category, it would be beneficial to list all six of these and a more specific reason for their exclusions.

One minor comment: the authors use the term "ROAUC". While this makes sense, it is a less common term than the more standard "AUROC". Use of the more common term would likely improve the paper's readability.

6. PLOS authors have the option to publish the peer review history of their article (what does this mean?). If published, this will include your full peer review and any attached files.

Reviewer #1: No

Reviewer #2: No

---

## [Author Response · Author response to Decision Letter 0]

21 Dec 2020

Response to Editor

Thank you for reviewing our manuscript entitled “Machine learning-based prediction of in-hospital mortality using admission laboratory data.” We are pleased to resubmit our revised manuscript to be considered for publication in PLOS ONE as an original paper.

We have revised the manuscript according to the reviewers’ valuable suggestions. One of the comments from Reviewer #1 suggested to clearly describe a baseline model and the advantage of the gradient descent boosting model over logistic regression model. In accordance with this suggestion, we have added a clear description of a baseline model, the logistic regression model, in our study. Additionally, using a calibration plot, we showed that the mortality risk of high-risk patients tended to be predicted accurately in the case of the gradient boosting decision tree. Another comment from Reviewer #2 required us to discuss the difference between the mortality in the test set and the training set. We visualized year-by-year mortality in S1 Fig and we discussed the prediction task using past data to make predictions in supervised learning. These instructions revealed the future challenges of this type of study, which were added as limitations to the Discussion section. Because detecting high-risk patients at the time of hospitalization leads to appropriately determining the medical resources needed according to the patient’s severity, we believe that our study is of general interest to PLOS ONE readers. In addition, we have revised the manuscript to address the editor’s comments as follows.

Thank you for these instructions. We have adjusted the author affiliations and headings in the manuscript accordingly, and lines 5-13, 19, 44, 69, 70, 104, 118, 158, 241, 309, 316, 323, and 431 of the manuscript have been revised.

Thank you for these instructions. The data used in this study were not openly available due to the restriction imposed by the research ethics committee of the Graduate School of Medicine and Faculty of Medicine, The University of Tokyo, because they contain sensitive patient information. Disclosure of data is not included in the ethics application, nor is it allowed in this study by hospital policy.

Therefore, we added the contact information of the research ethics committee to the manuscript according to the description in “Human research participant data and other sensitive data” under the Acceptable Data Access Restrictions (https://journals.plos.org/plosone/s/data-availability). Lines 74-78 of the manuscript have been revised accordingly. Specific contact information is as follows.

Office for Human Research Studies (OHRS): Graduate School of Medicine and Faculty of Medicine, The University of Tokyo, Faculty of Medicine Bldg. 2 4F, 7-3-1, Hongo, Bunkyo-ku, Tokyo, 113-0033, JAPAN, https://u-tokyo-ohrs.jp/en/

'This research was partially supported by the Tokyo Society of Medical Sciences and Grant-in-Aid for Scientific Research (C). The funders had no role in study design, data collection and analysis, decision to publish, or preparation of the manuscript.'

a. Please provide an amended statement that declares *all* the funding or sources of support (whether external or internal to your organization) received during this study, as detailed online in our guide for authors at http://journals.plos.org/plosone/s/submit-now

Please also include the statement “There was no additional external funding received for this study.” in your updated Funding Statement.

Thank you for these instructions. We have added the funding statement accordingly to the cover letter.

'I have read the journal's policy and the authors of this manuscript have the following competing interests: Y.K belongs to the 'Artificial Intelligence in Healthcare, Graduate School of Medicine, The University of Tokyo' which is an endowment department, supported with an unrestricted grant from ‘I&H Co., Ltd.’ and ‘EM SYSTEMS company’, but these sponsors had no control over the interpretation, writing, or publication of this work.'

a. Please confirm that this does not alter your adherence to all PLOS ONE policies on sharing data and materials, by including the following statement: "This does not alter our adherence to PLOS ONE policies on sharing data and materials.” (as detailed online in our guide for authors http://journals.plos.org/plosone/s/competing-interests). If there are restrictions on sharing of data and/or materials, please state these.

Please note that we cannot proceed with consideration of your article until this information has been declared.

Thank you for these instructions. We have added the description about competing interests to the cover letter.

5. Please include a copy of Table 1 which you refer to in your text on Line 156.

Thank you for pointing this out. The mention of Table 1 was a writing error and was meant to reference S1 Table. Lines 160 of the manuscript have been revised accordingly.

Response to Reviewer #1:

Thank you for reviewing the manuscript we submitted to PLOS ONE and for your valuable and educational comments. Based on your suggestions, we have revised the manuscript as described below.

Reviewer #1: The authors developed four machine learning models to predict in-hospital mortality risks. The models had similar performance in terms of discrimination and calibration. I have the following comments:

- The authors argued that they chose the lab data because it is the most manageable to input into the model. However, nowadays all data available in electronic health records can be readily input into a model that is integrated in the system. They might obtain better results if including additional data but also need to investigate whether the lab data does provide the majority of the prediction capability.

Thank you for your valuable comments and for pointing out these issues. As you indicated, data other than the laboratory data from electronic health records may be available for prediction model when the data structuring process is done properly. At this time in Japan, it is standard procedure to use SS-MIX2 standardized storage for storing medical data. This storage includes HL7 message files under the hierarchical structure folder. Laboratory data in this storage has become the main structured information to indicate patient state on admission and highly reproducible observation results that cannot be judged by humans. The data of the inspection department of the University of Tokyo Hospital are certified by ISO15189, which guarantees reproducibility of the results of the examination. 

As another example, this storage also includes prescription information as structured information. Prescription information is the result of human judgment; therefore, it may be biased by the judgment of individual doctors and are not unified explanatory variables. Additionally, in the case of first-visit patients, the lack of prescription information does not necessarily mean that there is no ongoing medication. To emphasize the reproducibility of explanatory variables, we narrowed down the data to laboratory data. 

As you indicated, unstructured information outside this standardized storage may show better results if included in explanatory variables after data structuring. However, shaping unstructured medical data into machine-readable information is very complicated and could not be achieved in this study. Therefore, we added it to the Discussion as a limitation to this study and mentioned that our results did not necessarily guarantee that laboratory data accounted for the majority of the prediction capability. Lines 254-262 of the manuscript have been revised accordingly.

- It is not clear what they meant by "variables not suitable for developing prediction model".

Thank you for pointing out this issue. The number of items used during preprocessing was incorrect and has been corrected. Three items were removed at this stage. Specifically, variables not suitable for developing a prediction model include the anonymous patient ID, date and time of hospitalization, and alkaline phosphatase value displayed in King-Armstrong unit. Because alkaline phosphatase values in IU/l were included in the data, we removed the values with King-Armstrong unit. Fig 1 and lines 93-97 of the manuscript have been revised accordingly.

- It is not mentioned in the paper, but I assume they performed multiple imputation for the test set also, judging from their figures. How did they calculate the predictions from the multiply imputed test set then? Please specify. Also, for application in a clinical setting, how do they expect the data to handle missingness?

Thank you for pointing out this issue. The prediction results were collected based on majority voting by aggregating the results of the imputation. While checking, we discovered that the confidence intervals for AUROC and AUPRC were based on the data after multiple substitutions without majority voting by aggregating the results of the imputation, so we recalculated the bootstrapped confidence interval based on 2000 replications using the prediction results collected based on majority voting by aggregating the results of the imputation. We have described this issue in the manuscript and lines 38-39, 110-116, and 167-169 of the manuscript have been revised.

As you have indicated, our models were based on multiple imputations for comparing machine learning methods that do not allow missing values. Because the laboratory variables used in this study were limited to variables with ≤ 25% missing data, rare laboratory tests were not included. However, it is an important limitation that these models do not admit missing values as input. We have added this limitation to the Discussion section and lines 277-282 of the manuscript have been revised.

- Why did the authors use the five models trained in training-fold of the cross-validation for application on the test set? The models can benefit more by being trained on the entire training set (or in the authors' terms, the training+validation data).

Thank you for pointing out this issue. In the model training methods used in this study, validation data were used for calibrating the trained model. If the entire training set (the training + validation data) were used for model training, then model calibration would become impossible. Additionally, in the case of the multilayer perceptron, validation data were used to decide the termination point of training using the early stopping protocol. Therefore, we avoided training on the entire training set at the test stage and selected evaluating assemble models derived from the test set cross-validation. We have added a description of this protocol and lines 145-146 and 151-155 of the manuscript have been revised.

- The models were calibrated during training phase. Were the models calibrated for the test set? There is discrepancy in the event rate between the training and test sets, which may require calibration.

Thank you for pointing out this issue. To imitate the evaluation of predictive performance for unknown data, model calibration was performed using validation data. In the actual use of this model, the discrepancy in the event rate between the available data and future data cannot be calibrated. Therefore, the models were calibrated on the validation set and evaluated on the test set. We have added a description of this discussion and lines 151-155 and 301-307 of the manuscript have been revised.

- For the calibration of the models, the authors need to show their results in deciles of risks or plot a smooth calibration curve. Right now the majority of patients are in the <1% risk category and it is hard to determine how accurately their risks are calibrated.

Thank you for pointing out this issue. In accordance with your suggestion, we have added the calibration plot derived from the results of the models performed on the test data set. Because the majority of patients are in the low-risk category, as you indicated, we added two calibration plots for each method. One calibration plot figure showed observed probability and predicted probability based on 10 quantiles of predicted probability and the other showed it based on 10 equal parts of predicted probability. We have added S2 Fig and a description of it, and lines 186-189 and 434-438 of the manuscript have been revised.

- Most importantly, the paper lacks a baseline model and has not explained the advantage of the gradient descent boosting model over logistic regression model. The authors need to compare their models with existing models on the same dataset; comparing roc-aucs on different datasets may not be fair. Also, the authors stated that the gradient descent boosting model had the best performance, but the difference from other models is small. It may be helpful to show percentage of patients who had better predictions.

Thank you for pointing out this issue. In accordance with your suggestion, we have clarified the logistic regression as the baseline model that uses linear feature extraction and the others as machine learning method with nonlinear feature extraction. Additionally, we deleted the section comparing AUROCs with another study using different datasets. To visualize the difference in the risk prediction of each model, a calibration plot based on 10 equal parts of predicted probability is shown in S2 Fig. We have added S2 Fig and a description of this point and lines 63-67, 186-189, 191-193, and 434-438 of the manuscript have been revised.

- The figures have very low resolution. The words in the figures are not legible.

Thank you for pointing out this issue. In accordance with your suggestion, we have corrected the characters in the figures to make them easier to read. Fig 1, Fig 2, Fig 3 and Fig 4 have been revised.

Response to Reviewer #2:

Thank you for reviewing our manuscript submitted to PLOS ONE and for your valuable and educational comments. Based on your suggestions, we have revised the manuscript as described below.

Reviewer #2: Seki ea al. present here a retrospective analysis of predicting mortality following a hospital stay by using several machine learning techniques in a dataset spanning nine years, 80k patients, and 174k admissions. They robustly utilized multiple imputation on missing variables, oversampled to account for outcome class imbalance, and clearly describe their approaches in grid searching model parameters. The model here uses 21 common lab results, age, and sex to predict mortality. As such, this is a model that could likely be applied widely. The authors note that a key limitation is that this model at present has only been applied to patients from a single hospital. I hope that in future work, this model is able to be tested and deployed more widely to aid in clinical care.

- One concern I have in reading this manuscript is that the mortality in the test set is so much lower than in the train set (0.69 vs 0.84). While the analysis and metrics used are not substantially impacted by this imbalance, it gives me concern that the model might not be optimally calibrated. What does year-by-year mortality look like in this dataset? Has it been increasing across the entire dataset, or is the decrease present only in the two years of the training set? If there are changes over time in the dataset, what impact would there be from removing an older year of training data? I realize that not all of these questions can be answered, but it would be good to see the authors acknowledge and discuss some of the implications of this change over time.

Thank you for pointing out this issue. In accordance with your suggestion, we have visualized year-by-year mortality in S1 Fig. Although mortality in 2016 and 2017 was comparatively low, it is not clear if there is a specific trend throughout the entire dataset. When predicting mortality with a machine learning model, using past data to predict future cases is a common method in supervised learning. In this study, to imitate the evaluation of predictive performance for unknown data, model calibration was performed using validation data. In the actual use of this model, the discrepancy in the event rate between the available data and future data cannot be calibrated. However, as you indicated, older data may affect model performance. Therefore, ongoing model updates are required for applying the machine learning model to clinical use. We have added a description of this limitation of machine learning models and lines 162, 301-307, and 432-433 of the manuscript have been revised.

- Throughout the paper, the authors reference different numbers of variables- either 25 or 23. I believe that the authors were uniform in their approach and that the 24th and 25th variables are length of stay and mortality, and so are not included as features. However, discussion such as in line 98 or in Figure 1 implies (perhaps misleadingly) that there were 25 features in the dataset. Clarification here would be useful.

Thank you for pointing out this issue. As you indicated, the 25 items are specifically age, sex, 21 laboratory variables, length of stay, and mortality. In accordance with your suggestion, we have added a description about this point and lines 93-97 of the manuscript have been revised.

- On lines 96-97, the authors note that variables "not suitable for developing the prediction model . . . were excluded." It would be helpful if the authors could clarify how these variables were unsuitable- for instance, is it because they were personal identification variables? Free text variables? Some other reason? As it appears from Figure 1 that 6 variables fall into this category, it would be beneficial to list all six of these and a more specific reason for their exclusions.

Thank you for pointing out this issue. The number of items used during preprocessing was incorrect and has been corrected. Three items were removed at this stage. Specifically, variables not suitable for developing prediction model include the anonymous patient ID, hospitalization time, and alkaline phosphatase value displayed in King-Armstrong unit. Because alkaline phosphatase values in IU/l were included in the data, we removed the values with King-Armstrong unit. We have added a description of this and Fig 1 and lines 93-97 of the manuscript have been revised.

- One minor comment: the authors use the term "ROAUC". While this makes sense, it is a less common term than the more standard "AUROC". Use of the more common term would likely improve the paper's readability.

Thank you for pointing out this issue. As you indicated, we replaced ROAUC with AUROC and PRAUC with AUPRC. In accordance with your suggestion, lines 36, 38, 130, 150, 155, 165, 167, 182, 184, 185, 186, 197, 198, 200, 201, 246, 197, 198, and 242 of the manuscript have been revised.

---

## [Decision Letter · Decision Letter 1]

25 Jan 2021

Machine learning-based prediction of in-hospital mortality using admission laboratory data: A retrospective, single-site study using electronic health record data

PONE-D-20-20161R1

Dear Dr. Seki,

We’re pleased to inform you that your manuscript has been judged scientifically suitable for publication and will be formally accepted for publication once it meets all outstanding technical requirements.

Kind regards,

Bobak Mortazavi

Academic Editor

PLOS ONE

Additional Editor Comments (optional):

Reviewers' comments:

Reviewer's Responses to Questions

**Comments to the Author**

1. If the authors have adequately addressed your comments raised in a previous round of review and you feel that this manuscript is now acceptable for publication, you may indicate that here to bypass the “Comments to the Author” section, enter your conflict of interest statement in the “Confidential to Editor” section, and submit your "Accept" recommendation.

Reviewer #1: All comments have been addressed

Reviewer #2: All comments have been addressed

2. Is the manuscript technically sound, and do the data support the conclusions?

Reviewer #1: (No Response)

Reviewer #2: (No Response)

3. Has the statistical analysis been performed appropriately and rigorously? 

Reviewer #1: (No Response)

Reviewer #2: (No Response)

4. Have the authors made all data underlying the findings in their manuscript fully available?

Reviewer #1: (No Response)

Reviewer #2: (No Response)

5. Is the manuscript presented in an intelligible fashion and written in standard English?

Reviewer #1: (No Response)

Reviewer #2: (No Response)

6. Review Comments to the Author

Reviewer #1: (No Response)

Reviewer #2: (No Response)

7. PLOS authors have the option to publish the peer review history of their article (what does this mean?). If published, this will include your full peer review and any attached files.

Reviewer #1: No

Reviewer #2: No

---

## [Editor Report · Acceptance letter]

27 Jan 2021

PONE-D-20-20161R1 

Machine learning-based prediction of in-hospital mortality using admission laboratory data: A retrospective, single-site study using electronic health record data 

Dear Dr. Seki:

I'm pleased to inform you that your manuscript has been deemed suitable for publication in PLOS ONE. Congratulations! Your manuscript is now with our production department. 

Kind regards, 

on behalf of

Dr. Bobak Mortazavi 

Academic Editor

PLOS ONE